# Health professionals' experiences and views on obstetric ultrasound in Vietnam: a regional, cross-sectional study

Sophia Holmlund,[1] Pham Thi Lan,[2] Kristina Edvardsson,[1,3] Ho Dang Phuc,[4] Joseph Ntaganira,[5] Rhonda Small,[3,6] Hussein Kidanto,[7] Matilda Ngarina,[8] Ingrid Mogren[3]

For numbered affiliations see end of article.

**Correspondence to**
Sophia Holmlund;
sofia.holmlund@umu.se

## ABSTRACT

**Objectives** Obstetric ultrasound is an important part of antenatal care in Vietnam, although there are great differences in access to antenatal care and ultrasound services across the country. The aim of this study was to explore Vietnamese health professionals' experiences and views of obstetric ultrasound in relation to clinical management, resources and skills.

**Design** A cross-sectional questionnaire study was performed as part of the CROss Country UltraSound study.

**Setting** Health facilities (n=29) in urban, semiurban and rural areas of Hanoi region in Vietnam.

**Participants** Participants were 289 obstetricians/gynaecologists and 535 midwives.

**Results** A majority (88%) of participants agreed that 'every woman should undergo ultrasound examination' during pregnancy to determine gestational age. Participants reported an average of six ultrasound examinations as medically indicated during an uncomplicated pregnancy. Access to ultrasound at participants' workplaces was reported as always available regardless of health facility level. Most participants performing ultrasound reported high-level skills for fetal heart rate examination (70%), whereas few (23%) reported being skilled in examination of the anatomy of the fetal heart. Insufficient ultrasound training leading to suboptimal pregnancy management was reported by 37% of all participants. 'Better quality of ultrasound machines', 'more physicians trained in ultrasound' and 'more training for health professionals currently performing ultrasound' were reported as ways to improve the utilisation of ultrasound.

**Conclusions** Obstetric ultrasound is used as an integral part of antenatal care at all selected health facility levels in the region of Hanoi, and access was reported as high. However, reports of insufficient ultrasound training resulting in suboptimal pregnancy management indicate a need for additional training of ultrasound operators to improve utilisation of ultrasound.

## BACKGROUND

Sufficient antenatal care (ANC) services and skilled birth attendance are important factors contributing to safer deliveries, reductions in obstetric complications, and

## Strengths and limitations of this study

► The study questionnaire was developed based on the results from earlier qualitative studies performed with obstetricians/physicians and midwives/nurses in six different countries (the CROss Country UltraSound study).

► The strengths of this study include participants of different health professional categories recruited from different levels of the healthcare system in urban, semiurban and rural areas of Hanoi.

► The research team comprised two Vietnamese researchers familiar with the setting and the healthcare system, which strengthens the interpretation of data.

► One limitation of this study may be the translation of the questionnaire from English to Vietnamese; however, measures to reduce the risk of losing the intended meaning of questions and statements were implemented.

► Since previous studies within this research domain are lacking, the power calculation was based on assumptions of proportions for one outcome variable in relation to one background variable, and may therefore mean uncertainty of the required study sample.

decreased maternal and neonatal morbidity and mortality.[1] Timely and appropriate evidence-based practices in ANC including health promotion, screening and diagnosis, and prevention for diseases can save lives.[2] Since 2016, WHO recommends a minimum of eight ANC contacts during pregnancy and one ultrasound examination before 24 weeks of gestation. The aim of the recommendation of an early ultrasound scan is to estimate gestational age, improve detection of multiple pregnancy and fetal anomaly, reduce induction of post-term pregnancy, and improve women's pregnancy experience.[2] Ultrasound is an important part of ANC in high-income

BMJ

countries,[3] and clinical trials show that ultrasound may improve management and pregnancy outcomes in the developing world.[4]

The utilisation of reproductive health services in Vietnam has substantially increased over recent decades, but there are still inequities in the country.[5] Low education, poverty, ethnic minority status[6] and living in rural areas are factors associated with decreased access to reproductive health services.[5] Ultrasound has become a central tool in ANC services in Vietnam.[7] Currently the Ministry of Health in Vietnam recommends at least four ANC visits,[8] in accordance with the previous recommendation by WHO.[9] Additionally, three ultrasound examinations are recommended during pregnancy, in gestational weeks 11–13, 20–24 and 30–32.[8] In 2014, more than 70% of pregnant women in Vietnam received at least four ANC visits.[10] However, almost half of these women reported not having urine or blood samples taken or blood pressure measurement done during their last pregnancy.[10] The private health sector is a continuously growing part of healthcare services in Vietnam, especially for provision of legal abortion services and obstetric ultrasound examinations.[11] Limited salary for physicians creates incentives for both public and private practitioners to search for additional income through provision of these services.[12 13] Commercialisation of ultrasound services has led to urban Vietnamese women having an average of more than six ultrasound scans during pregnancy, which is a high number of examinations from an international perspective.[12 14]

### Study rationale

Few studies have investigated the use of obstetric ultrasound in Vietnam from health professionals' perspectives, although ultrasound is frequently used during pregnancy. This study serves to fill a knowledge gap and can benefit authorities in their work to further develop education and improvement of guidelines on obstetric ultrasound use. This study is part of the CROss Country UltraSound study (CROCUS) investigating health professionals' experiences and views of the use of ultrasound in high-resource, middle-resource and low-resource countries.

### Aims

The overall aim of this study was to explore different aspects of obstetric ultrasound in Vietnam from health professionals' perspectives.

The following were the research questions investigated:

► What are health professionals' views of the role of obstetric ultrasound for clinical management of pregnancy?
► How do health professionals view access to obstetric ultrasound?
► How do health professionals assess their skills in performing obstetric ultrasound examinations?
► What do health professionals believe could improve the utilisation of obstetric ultrasound?

## MATERIALS AND METHODS

### The Vietnamese setting

Since 2010, Vietnam has been classified as a lower middle-income country and has undergone substantial economic development in recent decades.[1] The maternal mortality rate has decreased from 139/100 000 in 1990 to 54/100 000 live births in 2015.[15] Most inpatient healthcare is provided by public hospitals, but for outpatient care private clinics account for a large number of patients.[16] Vietnam is divided into 63 provinces, 698 districts and 11 121 communes.[17] In each commune, a village health worker (VHW) provides health promotion, immunisation and nutrition services, and attends births in remote areas.[18] At the community health centre level, a midwife or an assistant doctor is in charge of maternal health services, and provides ANC, assists normal delivery, and provides postnatal care, immunisation services and supervision of the VHW. At the district level, ANC, delivery care including caesarean sections and newborn care are provided at hospitals,[18] while maternity homes deliver basic prenatal and delivery services.[16 18] Provincial hospitals provide more specialised healthcare, and referrals from lower healthcare levels to provincial level are undertaken if complications occur during pregnancy or delivery.[18] At the top of the healthcare system, there are several national hospitals providing specialised care and receiving referrals from lower levels.[16]

### Study design

This cross-sectional study used a questionnaire to investigate a number of research questions related to obstetric ultrasound with obstetricians/gynaecologists and midwives providing pregnancy, delivery and postpartum care to women in the region of Hanoi, Vietnam.

### Sampling

Owing to the lack of findings from similar studies, a sample size of 290 obstetricians/gynaecologists and a corresponding number of midwives (n=290) was calculated based on plausible estimations of prevalence of background characteristics and outcome variables. The calculation was based on the outcome requiring the largest sample size, 'every woman should undergo ultrasound examination in pregnancy to determine gestational age', and the background variable 'work experience over and under 5 years', to detect a difference in proportion of 0.10 with a power of 80% and a significance level of 5%.

Purposive sampling was used to obtain a representative sample of health professionals caring for pregnant women at different levels of health facilities in urban, semiurban and rural areas in the region of Hanoi. One national hospital, 1 provincial hospital, 24 district hospitals and 3 maternity homes were included in the study, for a total of 29 health facilities.

### Questionnaire

The study questionnaire was developed based on the results from the earlier qualitative studies performed in

the CROCUS study.[7 19–27] Sociodemographic characteristics, evaluation of self-reported skills in performing ultrasound, and questions about access to obstetric ultrasound and health professionals' views on what may improve utilisation of ultrasound in Vietnam were included, among other items. The questionnaire included both questions and statements, and the items had either fixed or Likert-scale response options. This analysis investigates the research questions noted above, using 45 of the 105 questions and statements. The questionnaire was developed first in English and thereafter translated to Vietnamese by a native Vietnamese speaker independent of the research team. The Vietnamese version of the questionnaire was also back-translated to English by another external person. This check resulted in minor adjustments of some words, but demonstrated that the Vietnamese translation had retained the overall meaning of the English. The questionnaire was pilot-tested with 10 obstetricians, 6 midwives and 2 sonographers. No further revisions of the questionnaire were required as a result of piloting.

## Data collection procedures, including recruitment of study participants

The data collection was performed in April 2017 by four experienced data collectors supervised by two Vietnamese senior researchers in the research team. Before the start of the data collection, data collectors were trained by the research team, and all questions and statements in the questionnaire were discussed to ensure correct understanding. The two Vietnamese researchers initiated contact with the directors of all selected health facilities and all of them agreed to assist with recruitment of participants. For this study, we aimed to include health professionals caring for pregnant women and with different experiences in relation to use of obstetric ultrasound. Eligible participants were health professionals managing pregnant women at the maternity wards on the day of data collection at each study site. No eligible participant declined participation in the study. The primary sample included 890 participants. Six individuals working as radiology technicians were excluded from the primary sample as the they did not fulfil the inclusion criteria, and finally 60 sonographers were also excluded from the primary sample since they constituted a small part of the sample, and further did not contribute to clinical management after their obstetric ultrasound examination. The final sample (N=824) included the following health professionals: obstetricians/gynaecologists (n=289) and midwives (n=535). Participation was anonymous and all questionnaires were given a unique code. Safe storage of questionnaires was undertaken in accordance with national procedures and regulations. Data were entered into an SPSS file at Hanoi Medical University, by two experienced data clerks. To evaluate the quality of the data entry, every 10th questionnaire based on the number order of identification codes was selected for data re-entry. The data from all 107 variables in 89 questionnaires were re-entered in the SPSS file by the first author. The rate of error was 1.4%. The identified errors in the SPSS file were corrected.

## Patient and public involvement

This research was done without patient involvement.

## Independent variables

*Age* was calculated as a continuous variable using birth year and year of data collection. For some analyses age was dichotomised as age 34 years or less and 35 years and above. *Gender* included female or male. *Health profession* included the following response options on the questionnaire: obstetrician/gynaecologist, general practitioner, resident physician, physician other (please specify), midwife, radiologist/sonographer and 'other' (please specify). *Health profession* was thereafter categorised into two groups: obstetricians/gynaecologists and midwives. Resident physicians undergoing postgraduate training (n=9) and general practitioners (n=12) were also included in the category obstetricians/gynaecologists because they worked at the same department and performed similar work tasks as the obstetricians/gynaecologists. One participant who was an anaesthesiologist by profession but was working with maternity care was categorised as an obstetrician/gynaecologist. One nurse working in maternity care was categorised as a midwife. *Health facilities* included the response options national hospital, provincial hospital, district hospital and maternity home. The variable *health facilities* was dichotomised into national hospital/provincial hospital and district hospital/maternity home in some analyses. *Area of health facility* was categorised as hospitals in urban (n=7), semi-urban (n=5) and rural (n=17) areas of Hanoi. *Type of healthcare* was classified as public, private, and both public and private healthcare. No participant reported working only in private healthcare. *Number of ultrasound examinations indicated in an uncomplicated pregnancy* was categorised based on the three recommended number of ultrasound examinations by the Ministry of Health in Vietnam; three examinations or less and four examinations or more.

## Dependent variables

The dependent variables with fixed response alternatives are presented in box 1. For the statements related to 'the role of ultrasound for clinical management' and 'resources and training of obstetric ultrasound', the response options were dichotomised into *disagree* or *strongly disagree* and *agree* or *strongly agree* in logistic regression analyses. The question 'do you have a role in decision-making regarding clinical management on the basis of obstetric ultrasound examinations' was used both as an independent and dependent variable, and the response options were dichotomised into *no* and *yes* for some analyses. The response options for the statements related to 'improving utilisation of ultrasound' were categorised as *not at all* or *not very much* and *a fair amount* or *a great deal* in logistic regression analyses. The response option *neutral* or *don't know* was not included in either of these

**Box 1    Questions or statements and their response options in the questionnaire**

**How often do you perform obstetric ultrasound examinations?***
How do you rate your skills in ultrasound in relation to the assessment/evaluation of:
► Fetal presentation.†
► Localisation of the placenta.†
► Fetal heart rate.†
► Amniotic fluid amount.†
► Gestational age estimated by CRL (crown-rump-length).†
► Gestational age estimated by biparietal diameter, femur length and abdominal diameter.†
► Cervical length.†
► Fetal heart: four-chamber view.†
► Fetal heart: aorta and pulmonary artery.†
► Doppler: umbilical artery.†

**Do you have a role in decision-making regarding clinical management on the basis of obstetric ultrasound examinations?‡**

**How often do you make decisions based on the results from obstetric ultrasound examinations in your clinical work?***

**What do you believe would improve the utilisation of ultrasound at your clinic/workplace?**
► More ultrasound machines.§
► Better quality of ultrasound machines.§
► More training for health professionals currently performing ultrasound.§
► More doctors trained in ultrasound.§
► (More) midwives trained in ultrasound.§

**Statements on ultrasound resources and training.**
► Pregnant women in my country have access to dating ultrasound (ie, estimation of gestational age).¶
► Pregnant women in my country have access to fetal anomaly screening.¶
► Pregnant women in my country have access to obstetric ultrasound independent of area of living.¶
► Pregnant women in my country have access to obstetric ultrasound independent of income.¶
► There are enough resources in my country to provide *medically indicated* obstetric ultrasound examinations to pregnant women who need it.¶
► At my workplace, there is always access to obstetric ultrasound when it is needed.¶
► At my workplace, lack of ultrasound training of the ultrasound operator sometimes leads to suboptimal pregnancy management.¶
► Maternity care in my country would improve if midwives were qualified to perform basic ultrasound examinations.¶

**Statements on the role of ultrasound in clinical management of pregnancy.**
► Ultrasound is decisive in pregnancy management.¶
► Every woman should undergo ultrasound examination in pregnancy to determine gestational age.¶
► It is irresponsible of a pregnant woman to decline a dating scan.¶
► Ultrasound is safe to use for the pregnant woman and the fetus irrespective of the number of examinations.¶

Continued

**Box 1    Continued**

► Ultrasound is important for expectant parents to bond with their fetus during pregnancy.¶

*Response options: never, on a daily basis, on a weekly basis, on a monthly basis, more seldom than on a monthly basis.
†Response options: no skills, low skill-level, intermediate skill-level, high skill-level.
‡Response options: no, yes a minor role, yes a moderate role, yes a major role.
§Response options: not at all, not very much, a fair amount, a great deal, don't know.
¶Response options: strongly agree, agree, neutral, disagree, strongly disagree.

categories. For most statements, the response options *neutral* and *don't know* were selected by a small proportion of the participants.

## Statistical analysis

For categorical variables, frequencies and percentages were analysed and Pearson's $\chi^2$ test was used for test of difference, with the level of significance set at $p<0.05$. For continuous variables, mean values and their SDs were presented. Univariate and multivariable logistic regression was undertaken and presented with ORs and their 95% CIs. The independent and dependent variables used for logistic regression are reported in their specific sections as well as in box 1. All independent variables were entered into the univariate logistic regression analysis; however, only the statistically significant variables were included in the final multivariable logistic regression models. Statistical analyses were performed using SPSS V.23.

## RESULTS

### Background characteristics of the study sample

A total of 824 participants aged 21–60 years (mean age 34.8 years) were enrolled in the study. The distribution of health professionals was 35.1% obstetricians/gynaecologists and 64.9% midwives (table 1). One-third of the participants (28.1%) were performing obstetric ultrasound and mainly on a daily basis (66.5%). All obstetricians/gynaecologists working in maternity homes (100%) were performing ultrasound, and a majority of obstetricians/gynaecologists working in provincial hospitals (84.9%), district hospitals (75.2%) and national hospitals (75.7%). The mean estimated number of ultrasound examinations per day was 15.7 (median 10, range 1–100) for obstetricians/gynaecologists. Obstetricians/gynaecologists performing more than 10 examinations per day were significantly older (≥35 years) than those obstetricians/gynaecologists performing 10 or fewer examinations per day ($p<0.001$). A few participants at all healthcare levels (8.3%) reported that midwives performed ultrasound in their workplace. Participants at district hospitals (7.0%) and maternity homes (6.3%) reported the lowest percentage of midwives performing ultrasound in their workplace.

**Table 1** Background characteristics of the study sample (N=824)

| Variable | All health professionals Total=824 n (%) | Obstetricians/Gynaecologists Total=289 n (%) | Midwives Total=535 n (%) |
|---|---|---|---|
| Age (years) | 811 (98.4) | 286 (99.0) | 525 (98.1) |
| Mean; SD | 34.8; 8.7 | 36.6; 9.2 | 33.7; 8.3 |
| Minimum–maximum | 21–60 | 23–60 | 21–55 |
| Years in profession | 818 (99.3) | 288 (99.7) | 530 (99.1) |
| Mean; SD | 10.5; 8.3 | 10.4; 8.9 | 10.5; 8.1 |
| Minimum–maximum | 0–35 | 0–32 | 0.5–35 |
| Years in healthcare | 817 (99.2) | 287 (99.3) | 530 (99.1) |
| Mean; SD | 11.1; 8.5 | 11.6; 9.2 | 10.9; 8.2 |
| Minimum–maximum | 0–38 | 0–38 | 0.5–35 |
| Gender | 824 (100) | 289 (100) | 535 (100) |
| Male | 123 (14.9) | 123 (42.6) | 0 |
| Female | 701 (85.1) | 166 (57.4) | 535 (100.0) |
| Marital status | 817 (99.2) | 287 (99.3) | 530 (99.1) |
| Married | 714 (87.4) | 242 (84.3) | 472 (89.1) |
| Separated/divorced | 1 (0.1) | 0 (0.0) | 1 (0.2) |
| Widowed | 4 (0.5) | 2 (0.7) | 2 (0.4) |
| Not married/single | 98 (12.0) | 43 (15.0) | 55 (10.4) |
| Having children | 821 (99.6) | 288 (99.7) | 533 (99.6) |
| Yes | 684 (83.3) | 230 (79.9) | 454 (85.2) |
| No | 137 (16.7) | 58 (20.1) | 79 (14.8) |
| Type of healthcare | 823 (99.9) | 289 (100) | 534 (99.8) |
| Public | 789 (95.9) | 268 (92.7) | 521 (97.6) |
| Both public and private | 34 (4.1) | 21 (7.3) | 13 (2.4) |
| Level of health facility* | 824 (100) | 289 (100) | 535 (100) |
| National hospital | 144 (17.5) | 74 (25.6) | 70 (13.1) |
| Provincial hospital | 184 (22.3) | 86 (29.8) | 98 (18.3) |
| District hospital | 464 (56.3) | 121 (41.9) | 343 (64.1) |
| Maternity home | 32 (3.9) | 8 (2.8) | 24 (4.5) |
| Area of health facility† | 824 (100) | 289 (100) | 535 (100) |
| Urban | 439 (53.3) | 191 (66.1) | 248 (46.4) |
| Semiurban | 129 (15.7) | 35 (12.1) | 94 (17.6) |
| Rural | 256 (31.1) | 63 (21.8) | 193 (36.1) |
| Provision of maternity services‡ | | | |
| Antenatal care | 683 (83.0) | 261 (90.3) | 422 (79.0) |
| Intrapartum care | 642 (78.0) | 245 (84.8) | 397 (74.3) |
| Postpartum care | 688 (83.6) | 235 (81.3) | 453 (84.8) |
| Do not currently provide maternity care | 32 (3.9) | 10 (3.5) | 23 (4.3) |
| Performing ultrasound§ | 823 (99.9) | 289 (100) | 534 (99.8) |
| Yes | 231 (28.1) | 228 (78.9)¶ | 3 (0.6) |
| No | 592 (71.9) | 61 (21.1) | 531 (99.4) |

*Number of participants at specified health facilities.
†Number of participants at specified areas of health facilities.
‡Item on the questionnaire: 'Which of the following maternity services do you provide? (Please tick all that apply)'.
§Performing obstetric ultrasound examinations.
¶One participant has not rated the skills in relation to different tasks during ultrasound examinations.

### The role of obstetric ultrasound

Most participants (66.2%) agreed or strongly agreed that 'ultrasound is decisive in pregnancy management'. Obstetricians/gynaecologists reported significantly lower agreement (55.7%) with the statement 'ultrasound is decisive in pregnancy management' than midwives (71.8%; p<0.001). A majority (87.5%) of participants, independent of health profession, agreed or strongly agreed that 'every woman should undergo ultrasound examination to determine gestational age'. Most participants (75.0%) agreed or strongly agreed that 'it is irresponsible of a pregnant woman to decline a dating ultrasound', and there was no significant difference in opinion between different health professionals. There was a significant difference in opinion between the different health professionals for the statement 'ultrasound is safe to use for the woman and the fetus irrespective of the number of examinations', where obstetricians/gynaecologists were more likely to agree or strongly agree than midwives (OR 1.96; 95% CI 1.22 to 3.17). Participants reported an average of 5.9 ultrasound examinations as medically indicated during an uncomplicated pregnancy (obstetricians/gynaecologists: SD 2.7, range 2–15; midwives: SD 2.6, range 2–20). A quarter of the ultrasound operators (25.1%) agreed with the national guidelines that three ultrasound examinations are medically indicated during an uncomplicated pregnancy. Midwives, in comparison with obstetricians/gynaecologists, were more likely to agree or strongly agree that 'ultrasound is important for expectant parents to bond with their fetus during pregnancy' (OR 1.59; 95% CI 1.13 to 2.26). The assessment that four ultrasound examinations or more are medically indicated in an uncomplicated pregnancy was associated with higher agreement with the statement 'ultrasound is important for expectant parents to bond with their fetus during pregnancy', compared with those assessing that three ultrasound examinations or fewer are medically indicated in a normal pregnancy (OR 1.61; 95% CI 1.03 to 2.50, adjusted for health profession and performing ultrasound or not).

### Access to obstetric ultrasound

Most of the participants (95.6%–100%), regardless of health facility level, agreed or strongly agreed with the statement 'there is always access to obstetric ultrasound when needed at my workplace'. Almost all participants (95.4%–100%) at all health facility levels reported that they agreed or strongly agreed with the statement 'pregnant women in the country have access to dating ultrasound'. A majority of participants at all health facility levels (93.8%–95.1%) agreed or strongly agreed that 'pregnant women in the country have access to fetal anomaly screening'. Midwives were more likely to agree or strongly agree that 'pregnant women in the country have access to ultrasound independent of area of residence', compared with obstetricians/gynaecologists (OR 2.54; 95% CI 1.60 to 4.02). Participants in national hospitals and provincial hospitals reported significantly lower agreement with the statement 'pregnant women in my country have access to ultrasound independent of income' than participants in district hospitals and maternity homes (p<0.001). Further results are presented in table 2.

### Ultrasound operators' decision-making and self-rated skills

Almost all obstetricians/gynaecologists (92.2%) and a majority of midwives (59.4%) reported having a role in decision-making regarding clinical management on the basis of obstetric ultrasound examinations. There was a significant difference in the proportion of obstetricians/gynaecologists (38.6%) and midwives (20.5%) reporting having a major role in decision-making compared with having a minor or moderate role in decision-making (p<0.001). Approximately one-third of all health professionals (37.1%) reported that they agreed or strongly agreed with the statement 'at my workplace, lack of ultrasound training of the ultrasound operator sometimes leads to suboptimal pregnancy management'. Health professionals performing ultrasound were asked to rate their skills in relation to different tasks during ultrasound examinations (figure 1). For all the obstetricians/gynaecologists performing ultrasound (n=227), fetal heart rate was the examination where most participants reported high skills (70.0%), and examination of the fetal heart: aorta and pulmonary artery was the examination with the lowest proportion of self-rated high skill level (22.5%).

### Improving utilisation of obstetric ultrasound

When health professionals were asked to assess how much particular strategies could improve the utilisation of ultrasound, 'better quality of ultrasound machines' (94.0%), 'more training for health professionals currently performing ultrasound' (92.8%) and 'more physicians trained in ultrasound' (93.2%) were the statements with the highest numbers of participants reporting a fair amount or a great deal. Obstetricians/gynaecologists were more likely to agree that 'more training for health professionals currently performing ultrasound would improve the utilisation of ultrasound' compared with midwives (OR 2.60; 95% CI 1.24 to 5.46). Further results are presented in table 3. A majority of participants (69.0%) agreed or strongly agreed with the statement 'maternity care in my country would improve if midwives were qualified to perform basic ultrasound examinations'; however, midwives were more likely to agree or strongly agree than obstetricians/gynaecologists (OR 5.09; 95% CI 3.41 to 7.61). Participants working in rural hospitals (79.6%) and semiurban hospitals (78.3%) were also more likely to agree with the previous statement than participants working in urban hospitals (60.0%) (p<0.001).

### DISCUSSION

The main findings of this study are that access to obstetric ultrasound was generally reported as satisfactory regardless of participants' health profession and health facility level. Participants reported an average of almost six ultrasound

**Table 2** Health professionals' views on specified statements* (N=824)

| Statement | Obstetricians/ Gynaecologists Total=289† n (%) | Midwives Total=535† n (%) | P value‡ |
|---|---|---|---|
| Pregnant women in my country have access to dating ultrasound (ie, estimation of gestational age). | 279 (96.5) | 521 (97.4) | 0.98 |
| Agree/Strongly agree | 275 (98.6) | 512 (98.3) | |
| Disagree/Strongly disagree | 4 (1.4) | 9 (1.7) | |
| Pregnant women in my country have access to fetal anomaly screening. | 276 (95.5) | 514 (96.1) | 0.16 |
| Agree/Strongly agree | 269 (97.5) | 509 (99.0) | |
| Disagree/Strongly disagree | 7 (2.5) | 5 (1.0) | |
| Pregnant women in my country have access to obstetric ultrasound independent of area of residence. | 248 (85.8) | 485 (90.7) | <0.001 |
| Agree/Strongly agree | 203 (81.9) | 446 (92.0) | |
| Disagree/Strongly disagree | 45 (18.1) | 39 (8.0) | |
| Pregnant women in my country have access to obstetric ultrasound independent of income. | 228 (78.9) | 475 (88.8) | 0.082 |
| Agree/Strongly agree | 178 (78.1) | 398 (83.8) | |
| Disagree/Strongly disagree | 50 (21.9) | 77 (16.2) | |
| There are enough resources in my country to provide medically indicated obstetric ultrasound examinations to pregnant women who need it. | 234 (81.0) | 447 (83.6) | 0.002 |
| Agree/Strongly agree | 185 (79.1) | 395 (88.4) | |
| Disagree/Strongly disagree | 49 (20.9) | 52 (11.6) | |
| At my workplace, there is always access to obstetric ultrasound when it is needed. | 282 (97.6) | 519 (97.0) | 0.010 |
| Agree/Strongly agree | 272 (96.5) | 515 (99.2) | |
| Disagree/Strongly disagree | 10 (3.5) | 4 (0.8) | |
| At my workplace, lack of ultrasound training of the ultrasound operator sometimes leads to suboptimal pregnancy management. | 237 (82.0) | 452 (84.5) | 0.086 |
| Agree/Strongly agree | 115 (48.5) | 187 (41.4) | |
| Disagree/Strongly disagree | 122 (51.5) | 265 (58.6) | |

*The response option *neutral* was excluded from analyses.
†The number of participants included in each analysis is presented in relation to the total sample of each group of obstetricians/ gynaecologists and midwives.
‡Pearson's $\chi^2$ test with Yates' continuity correction for test of differences in proportion between the two groups of obstetricians/ gynaecologists and midwives.

examinations as medically indicated in an uncomplicated pregnancy in contrast to the three ultrasound examinations that are recommended in the Vietnamese national guidelines. Obstetricians/gynaecologists reported high self-rated skill levels for most obstetric ultrasound examinations, although one-third of all health professionals reported that 'lack of ultrasound training sometimes leads to suboptimal pregnancy management'. 'Better quality of ultrasound machines', 'more training for health professionals currently performing ultrasound' and 'more physicians trained in ultrasound' were reported as factors that could improve the utilisation of ultrasound to

the greatest extent. For participants at health facilities in rural and semiurban areas compared with those in urban areas, the great majority reported that 'maternity care in my country would improve if midwives were qualified to perform basic ultrasound examinations'.

### The role of obstetric ultrasound
Obstetric ultrasound is one of the most important technological advances in pregnancy surveillance.[28] The majority of health professionals in our study agreed or strongly agreed with the statement 'ultrasound is decisive in pregnancy management', but unexpectedly one-third

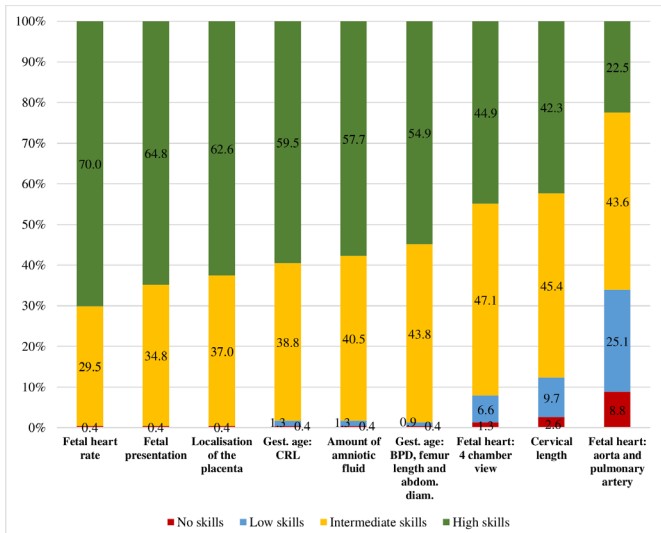

**Figure 1** Obstetricians/gynaecologists' self-rated skills for specified ultrasound examinations (n=227). Reported skill levels are presented with proportions. abdom. diam, abdominal diameter; BPD, biparietal diameter; CRL, crown-rump-length; gest., gestational.

of our participants were neutral or disagreed with this statement. Obstetricians/gynaecologists who mainly performed ultrasound reported lower agreement with the above statement than midwives who mainly did not perform ultrasound examinations. The findings of the CROCUS study in Rwanda also demonstrated that health professionals not themselves performing ultrasounds have more liberal attitudes towards ultrasound use than health professionals performing these examinations.[29] Health professionals sometimes fear that 'routinisation' of ultrasound as an unquestioned and integral part of pregnancy management can exert a negative impact on pregnant women's informed consent, and that its increasing use for fetal examination may have consequences for how disability is viewed in society.[25 26 30] The use of ultrasound may also reduce attention on important clinical parameters such as measurement of blood pressure and proteinuria.[7 24] Previous research from Vietnam has shown that many ultrasound examinations are performed just for reassurance of fetal well-being.[12] Although problems of access to obstetric ultrasound exist in many low-resource countries, inappropriate use of ultrasound examinations with no effect on pregnancy outcomes still occurs. In a study from Uganda, more than half of all ultrasound examinations performed were classified as inappropriate, for example dating of pregnancy in a suboptimal gestational week or requesting an ultrasound without medical indication.[31] The Ministry of Health in Vietnam recommends three ultrasound examinations including one examination also in the third trimester. However, a routine late pregnancy ultrasound in unselected or low-risk populations has been reported to have no benefit for the mother or the baby.[32] It is known that Vietnamese women generally undergo many ultrasound examinations during pregnancy.[12] Although it is well known that

ultrasound examinations hold a strong appeal for pregnant women,[33] the high number of examinations that pregnant women receive seems also to be supported by the ultrasound operators, as the participants in this study considered that twice as many ultrasound examinations were medically indicated compared with the recommendations by the Ministry of Health in Vietnam.[8] Further, participants' report of a mean number of almost six ultrasound examinations to be medically indicated is also in great contrast to the Cochrane review supporting the 2016 WHO ANC guideline recommending a single ultrasound examination before 24 weeks of gestation.[2 34] Health professionals performing ultrasound in our study also reported performing up to 100 scans per day per ultrasound operator. High workload can lead to inadequate provision of information to women by the physician, including about indication for ultrasound and the results of the ultrasound examination.[12 35] In a broader perspective, non-medical ultrasound examinations during pregnancy consume resources unnecessarily, with negative impact on other maternal healthcare services. Issuing medical guidelines stating clear indications for ultrasound surveillance during pregnancy is therefore important and may contribute to more appropriate allocation of resources within the healthcare system in Vietnam.

### Access to obstetric ultrasound
The health professionals in our study generally reported satisfactory access to obstetric ultrasound when needed in their own workplace. Access to ultrasound has increased significantly in many resource-limited settings, although there are still large differences in access within and between countries.[36] Our study was performed in the area around Hanoi, that is, the capital of Vietnam, and it is plausible to believe that access to obstetric ultrasound is higher in this region than in other parts of Vietnam. In Hanoi, ultrasound scans are easily accessible and affordable for most pregnant women both within the public and private healthcare systems. Increased ANC attendance and health facility delivery rates have been seen in Vietnam in recent years, but also increased inequities in maternity care utilisation,[37 38] primarily among women with multiple socioeconomic vulnerabilities.[37]

### Ultrasound operators' skills
Participants performing ultrasound in our study reported high or intermediate skill levels for the majority of the specified ultrasound examinations. Proper training of health professionals performing ultrasound is critically important,[28] especially since ultrasound is operator-dependent to a large extent.[4] Further, ultrasound training should include ethics and discuss use and misuse of the ultrasound tool, in addition to the quality of ultrasound performance and clinical implications of its use,[28] to ensure maximum diagnostic utility and high levels of sensitivity and specificity.[4]

**Table 3** Health professionals' views on factors that may improve utilisation of obstetric ultrasound, presented through preformed statements

| Variable | More ultrasound machines* | | | Better quality of ultrasound machines* | | |
|---|---|---|---|---|---|---|
| | Not at all or not very much† | A fair amount or a great deal† | | Not at all or not very much† | A fair amount or a great deal† | |
| | n (%) | n (%) | P value‡ | n (%) | n (%) | P value‡ |
| Health profession | 149 (20.1) | 593 (79.9) | 0.042 | 40 (5.2) | 732 (94.8) | 0.036 |
| Obstetricians/ gynaecologists | 65 (24.3) | 203 (75.7) | | 8 (2.8) | 276 (97.2) | |
| Midwives | 84 (17.7) | 390 (82.3) | | 32 (6.6) | 456 (93.4) | |
| Level of health facility | 149 (20.1) | 593 (79.9) | 0.003 | 40 (5.2) | 732 (94.8) | 0.232 |
| National hospital | 31 (24.4) | 96 (75.6) | | 3 (2.2) | 135 (97.8) | |
| Provincial hospital | 43 (28.3) | 109 (71.7) | | 12 (7.2) | 154 (92.8) | |
| District hospital | 73 (16.9) | 359 (83.1) | | 24 (5.5) | 413 (94.5) | |
| Maternity home | 2 (6.5) | 29 (93.5) | | 1 (3.2) | 30 (96.8) | |
| Performing ultrasound§ | 149 (20.1) | 592 (79.9) | 0.045 | 40 (5.2) | 731 (94.8) | 0.060 |
| Yes | 53 (25.0) | 159 (75.0) | | 6 (2.6) | 221 (97.4) | |
| No | 96 (18.1) | 433 (81.9) | | 34 (6.3) | 510 (93.8) | |

| | More training for health professionals currently performing ultrasound* | | | More physicians trained in ultrasound* | | |
|---|---|---|---|---|---|---|
| | Not at all or not very much† | A fair amount or a great deal† | | Not at all or not very much† | A fair amount or a great deal† | |
| | n (%) | n (%) | P value‡ | n (%) | n (%) | P value‡ |
| Health profession | 48 (6.3) | 717 (93.7) | 0.014 | 43 (5.7) | 714 (94.3) | 1.000 |
| Obstetricians/ gynaecologists | 9 (3.2) | 269 (96.8) | | 16 (5.7) | 263 (94.3) | |
| Midwives | 39 (8.0) | 448 (92.0) | | 27 (5.6) | 451 (94.4) | |
| Level of health facility | 48 (6.3) | 717 (93.7) | 0.097 | 43 (5.7) | 714 (94.3) | 0.047 |
| National hospital | 6 (4.5) | 126 (95.5) | | 3 (2.3) | 129 (97.7) | |
| Provincial hospital | 15 (9.1) | 149 (90.9) | | 15 (9.2) | 148 (90.8) | |
| District hospital | 23 (5.2) | 416 (94.8) | | 22 (5.1) | 410 (94.9) | |
| Maternity home | 4 (13.3) | 26 (86.7) | | 3 (10.0) | 27 (90.0) | |
| Performing ultrasound§ | 48 (6.3) | 716 (93.7) | 0.034 | 43 (5.7) | 713 (94.3) | 0.950 |
| Yes | 7 (3.2) | 215 (96.8) | | 12 (5.4) | 211 (94.6) | |
| No | 41 (7.6) | 501 (92.4) | | 31 (5.8) | 502 (94.2) | |

| | (More) midwives trained in ultrasound* | | |
|---|---|---|---|
| | Not at all or not very much† | A fair amount or a great deal† | |
| | n (%) | n (%) | P value‡ |
| Health profession | 277 (37.5) | 462 (62.5) | <0.001 |
| Obstetricians/ gynaecologists | 165 (60.7) | 107 (39.3) | |
| Midwives | 112 (24.0) | 355 (76.0) | |
| Level of health facility | 277 (37.5) | 462 (62.5) | 0.003 |
| National hospital | 50 (38.2) | 81 (61.8) | |
| Provincial hospital | 79 (49.4) | 81 (50.6) | |
| District hospital | 139 (33.4) | 277 (66.6) | |
| Maternity home | 9 (28.1) | 23 (71.9) | |
| Performing ultrasound§ | 277 (37.5) | 461 (62.5) | <0.001 |
| Yes | 135 (62.2) | 82 (37.8) | |
| No | 142 (27.3) | 379 (72.7) | |

Continued

**Table 3** Continued

| | (More) midwives trained in ultrasound* | | |
|---|---|---|---|
| | Not at all or not very much† | A fair amount or a great deal† | |
| | n (%) | n (%) | P value‡ |

*Item on the questionnaire: 'What do you believe would improve the utilisation of ultrasound at your clinic/work place?'
†The response option *don't know* was excluded from analyses.
‡Pearson's $\chi^2$ test for categorical variables. Yates' continuity correction was used for 2 by 2 tables.
§Performing obstetric ultrasound examinations.

In our study, one-third of all health professionals believed that insufficient training of the ultrasound operator sometimes leads to suboptimal pregnancy management. Another study from Vietnam reports that physicians experience less knowledge about fetal anomalies, lack advanced training, and do not have appropriate equipment and professional protocols that could support their practice in performing obstetric ultrasound.[39] Inadequately skilled health professionals may cause inadvertent harm, for example, by providing false-positive diagnosis where termination of pregnancy may be an option. Alternatively, false-negative information because of inadequate ability to recognise the signs of important diagnoses can result in parents not being offered further investigations and being inappropriately reassured that everything is normal.[40]

### Improving utilisation of ultrasound
Our results indicate an increased number of physicians to become appropriately educated in ultrasound examinations, regular inservice training sessions and better quality of ultrasound equipment. Lack of training of healthcare providers has been seen as the most common barrier to regular ultrasound use, although lack of equipment and maintenance and costs for machines also are explanatory factors.[41] A review of ultrasound training in low-income and middle-income countries (LMICs) shows that health professionals often do not meet the WHO criteria in relation to number of scans, supervision and length of training.[42] The obstacles to ultrasound training in LMICs include lack of time for training because of limited possibilities for absence from the workplace and the logistics to access qualified teachers.[42] Our results also indicate that health professionals in Vietnam seem to have a very substantial workload. It has been shown that training of midwives in basic obstetric ultrasound may significantly improve pregnancy management.[43–45] In our study, the majority of midwives were positive to the idea of training midwives in ultrasound to improve utilisation of ultrasound. For rural and semiurban hospitals, a majority of participants were also positive to ultrasound-trained midwives. However, obstetricians/gynaecologists did not support the idea of training midwives in ultrasound as much as participating midwives. As in much of Vietnamese society, health professions are organised hierarchically. Nurses' roles can be seen as primarily carrying out doctors' orders, not to take their own initiatives,[46 47]

and it may be a plausible explanation for why physicians defend performing ultrasound as their duty. Shortages of nurses and midwives and lack of quality of training[48 49] may also be additional explanations for physicians not supporting the idea of training midwives in obstetric ultrasound.

### Strengths and limitations
The strengths of this study include participants recruited from different levels of the healthcare system in urban, semiurban and rural areas of Hanoi. We believe that the wide range of health facilities is likely to be representative of the Vietnamese healthcare system. The research team comprised two Vietnamese researchers familiar with the setting and the healthcare system, which strengthens the interpretation of data. An additional strength was that four experienced Vietnamese data collectors, familiar with the setting, collected all data. One limitation of this study may be the translation of the questionnaire from English to Vietnamese, with the risk of losing the intended meaning of questions and statements. However, measures to reduce this risk were implemented, including back-translation of the questionnaire. Another limitation might be the unequal distribution between the categories of physicians and midwives in this sample. Although we aimed for equal numbers of each category of health professional, as assessed in the power calculation, all eligible physicians and midwives working the day of data collection were included in the study. This resulted in a higher number of midwives. Since there is a lack of previous studies within this research domain, the power calculation was based on assumptions of proportions for one outcome variable in relation to one background variable. The assumptions in the power calculations therefore mean uncertainty of the required study sample. However, the results indicate that the sample size was probably adequate for investigating our questions of interest. There was no question on the questionnaire about whether the participant had received any formal ultrasound training and this may be considered a limitation. In Vietnam, it is widely recognised that participants in studies receive payment. All participants were therefore paid with a small sum for answering the questionnaire, which may theoretically have affected the willingness of participation in the study.

## CONCLUSIONS

Obstetric ultrasound is used as an integral part of ANC at all selected health facility levels in the region of Hanoi, and access to obstetric ultrasound was also reported to be high at all levels. Overall, participants performing obstetric ultrasound reported satisfactory self-rated skill levels. However, report of insufficient ultrasound training resulting in suboptimal pregnancy management, in combination with the suggestion for more training to improve utilisation of ultrasound, indicates a need for additional inservice training of ultrasound operators. The proposal of educating midwives in obstetric ultrasound needs to be further evaluated.

**Author affiliations**
[1]Department of Clinical Sciences, Obstetrics and Gynecology, Umeå University, Umeå, Sweden
[2]Department of Dermatology and Venereology, Hanoi Medical University, Hanoi, Vietnam
[3]Judith Lumley Centre, School of Nursing and Midwifery, La Trobe University, Melbourne, Victoria, Australia
[4]Department of Probability and Mathematical Statistics, Institute of Mathematics, Vietnam Academy of Science and Technology, Hanoi, Vietnam
[5]School of Public Health, University of Rwanda, College of Medicine and Health Sciences, Kigali, Rwanda
[6]Department of Women's and Children's and Reproductive Health, Karolinska Institutet, Stockholm, Sweden
[7]Department of Obstetrics and Gynecology, Aga Khan University - Tanzania, Dar es Salaam, Tanzania
[8]Department of Obstetrics and Gynecology, Muhimbili National Hospital, Dar es Salaam, Tanzania

**Acknowledgements** Our sincere thanks to the participating health professionals for sharing their time and experiences and for the support by the heads of the Departments of Obstetrics and Gynaecology at the selected health facilities. Thanks go to the data collectors Do Nam Khanh, Nguyen Huyen Tram, Le Vu Thuy Huong and Nguyen Thi Hue for excellent performance in the data collection. We also thank Annika Åhman, Uppsala University, Sweden, for valuable contributions to the questionnaire, and Gabriel Granåsen, Umeå University, Sweden, for statistical support. We acknowledge the support received from Hanoi Medical University (Hanoi, Vietnam) and Umeå University (Umeå, Sweden).

**Contributors** IM is the principal investigator of the CROCUS study. SH, LPT, KE, JN, RS, HLK, MN and IM designed the study. SH, KE and IM were mainly responsible for developing the questionnaire, with input from all coauthors. LPT, PHD, SH, KE and IM organised and performed the data collection together with the data collectors. SH conducted the analyses in collaboration with IM, LPT and PHD. SH drafted the primary manuscript in collaboration with IM. All authors have contributed to and approved the final manuscript, tables and figure.

**Funding** This research was made possible by grants from Umeå University, Västerbotten County Council, the Swedish Research Council, Sweden (2014-2672), the Swedish Research Council for Health, Working Life and Welfare (Forte), and the European Commission under a COFAS Marie Curie Fellowship (2013-2699).

**Competing interests** None declared.

**Patient consent for publication** Not required.

**Ethics approval** Hanoi Medical University Review Board in Bio-Medical Research (reference 141/HMU IRB).

**Provenance and peer review** Not commissioned; externally peer reviewed.

**Data availability statement** Data are available upon reasonable request.

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
