## [Reviewer comments · BMJ Open]

ARTICLE DETAILS

TITLE (PROVISIONAL)	Health professionals' experiences and views on obstetric ultrasound in Vietnam: a regional, cross-sectional study
AUTHORS	Holmlund, Sophia; Pham Thi, Lan; Edvardsson, Kristina; Ho Dang, Phuc; Ntaganira, Joseph; Small, Rhonda; Lesio Kidanto, Hussein; Ngarina, Matilda; Mogren, Ingrid

VERSION 1 – REVIEW

REVIEWER	Torbjørn Moe Eggebø Center for fetal medicine Trondheim University Hospital Norway
REVIEW RETURNED	19-Jan-2019

GENERAL COMMENTS	Thank you for letting me review this manuscript. It is very interesting. Unfortunately, the manuscript is too long and overloaded with results. The method and result sections can be shortened. Especially, the text in the result section can be condensed. In general, the same results should not be presented both in tables and in text. Only, the most important findings should be repeated in the text. In accordance with author guidelines, five tables and figures are recommended. However, altogether nine is included in this study (six tables and three figures). As few as possible abbreviations should be used. The abbreviations of authors' contribution are not necessary to include in the text. The readers will not remember these abbreviations. The contribution to all authors should rather be presented as a paragraph in the end. The aim of the study was to explore different aspects of obstetric ultrasound in Vietnam from health professionals' perspectives. Vietnam is classified as a mid-resource country, and investigating the use of ultrasound in pregnancy is interesting. The authorities recommend three ultrasound examinations during pregnancy, and this is more than medically indicated. At what time in pregnancy is scanning recommended? The clinical value of a routine third trimester scan is not documented. In private sector, up to six examinations are performed in Vietnam. I miss a question about health professionals' view on the appropriate number of ultrasound examinations in an uncomplicated pregnancy. I am aware that this question was included in another part of the CROCUS study. How many questionnaires were distributed and what was the response rate? Four important main questions are presented, and the manuscript is well structured in accordance with these questions.
---

	“What are health professionals’ views of the role of obstetric ultrasound for clinical management of pregnancy? How do health professionals view access to obstetric ultrasound? How do health professionals assess their skills in performing obstetric ultrasound examinations? What do health professionals believe could improve the utilisation of obstetric ultrasound?” The introduction and discussion are well written and highlight the important findings. The figures are illustrative. The power calculation is difficult to understand, because it is based on two different questions. Please explain the power calculation better. In accordance with the power calculation 290 obstetricians and 290 midwives should participate in the study. It is not appropriate to include another group not included in the power analyses (58 sonographers) even though sonographers perform ultrasound examinations in Vietnam. The sonographers are only 6.6% of participants, and this group confuses the method section and the result presentation. In some statistical analyses they are a separate group and in other analyses they are combined with obstetricians (Table 3). This leads to multiple analyzing, and high risk of statistical type 1 error. Sonographers are sometimes called physicians (page 11, line 252) and in other sections doctors are called physicians (page 24, line 513). I suggest omitting this group in accordance with the power calculation. Why did you include 535 midwives when the power calculation showed 290 needed? The unequal distribution between doctors and midwives is a limitation which may have influenced results. Please discuss. In Table 3, four degrees of agreement are tested in three groups; leading to a cross table with 12 squares. It is still possible to use chi-square test, but not very interesting. The significance test only shows that differences between the extreme values exist. It is not appropriate to first analyse all three groups separately, and thereafter include sonographers in the group with obstetricians. The logistic regression analyses are not explained in the statistic section. Which variables were included, and how were they selected? Limitations related to including the small group of sonographers with all problems related to this are not adequately discussed. The conclusion should only present results from the study and not the authors meanings. The participants are not asked about guidelines, thus, the last sentence in conclusion should be omitted: “In addition, issuing medical guidelines stating clear indications for ultrasound surveillance during pregnancy may contribute to adequate allocation of resources within the health care system in Vietnam.” This study has a very high rate of “self-citing” (11/47 =23%). Please search carefully for relevant references from other groups.
--	---

REVIEWER	Ashraf Talaat Youssef Fayoum university Egypt
REVIEW RETURNED	07-Feb-2019

GENERAL COMMENTS	Reviewer comments : The article evaluated the views and experience of health professionals in obstetric ultrasound in Vietnam, the article was good written ,long with many tables and figures and discussed briefly the use of obstetric ultrasound as regard the limitations and how to improve the utility of ultrasound but some data needs to be explained. The authors mentioned that some operators may perform 100 exam a day ,which mean that the exam time could be <5 min if they work 8 hours continuously , Which is not enough to obtain useful data and the work overload is a factor that limits the proper use of ultrasound. Did the patients receive ultrasound reports ? . Did the operators have a certificate of practice ultrasound? Did they received a qualified ultrasound courses? And what are the requirements that should be fulfilled in the courses?as regards to time ,number of exams that they should attend and the ultrasound skills that they acquire?. Did the operators learn transvaginal ultrasound to assess early pregnancy and its complications ? .Did the radiologists have a role in obstetric ultrasound ? .Did the operators learn when to consult and for whom they refer difficult cases?.Why the authors asked the views and experience of many midwives while they lack experience with ultrasound.
--

REVIEWER	Hein Lamprecht Division of Emergency Medicine, Faculty of Medicine and Health Sciences, Stellenbosch University, South Africa
REVIEW RETURNED	07-Feb-2019

GENERAL COMMENTS	Although the manuscript is well written and complete (as supported by my review checklist), I am of the opinion that the research question and findings add nothing new to the current literature on the specific research topic. The utility of ultrasound in low resource settings have been published umpteenth times including the barriers to the implementation of ultrasound identified in this manuscript.
---

REVIEWER	Godofreda V. Dalmacion University of the Philippines Manila, Philippines (retired professor)
REVIEW RETURNED	24-Feb-2019

GENERAL COMMENTS	The reviewer provided a marked copy with additional comments. Please contact the publisher for full details.
---

VERSION 2 – AUTHOR RESPONSE

Response to Reviewers

Reviewer no 1		
Item	Reviewers comments	Authors' responses
1	Thank you for letting me review this manuscript. It is very interesting. Unfortunately, the manuscript is too	Thank you for your overall positive comments including your assessment that our manuscript is interesting!

	long and overloaded with results. The method and result sections can be shortened. Especially, the text in the result section can be condensed. In general, the same results should not be presented both in tables and in text. Only, the most important findings should be repeated in the text. In accordance with author guidelines, five tables and figures are recommended. However, altogether nine is included in this study (six tables and three figures).	In fact, the findings in our manuscript are novel for the Vietnamese setting. In accordance with your suggestions, two tables and two figures have been removed, and the result section have been condensed. As requested, text has been added to the Statistical analysis section (suggestion presented in item 11), as well as other sections when asked for by the reviewers.
2	As few as possible abbreviations should be used. The abbreviations of authors contribution are not necessary to include in the text. The readers will not remember these abbreviations. The contribution to all authors should rather be presented as a paragraph in the end.	Thank you for this comment. As suggested, the author abbreviation have been removed from the text. We agree that it is enough to have a paragraph at the end presenting author contributions (already in place).
3	The aim of the study was to explore different aspects of obstetric ultrasound in Vietnam from health professionals' perspectives. Vietnam is classified as a mid-resource country, and investigating the use of ultrasound in pregnancy is interesting. The authorities recommend three ultrasound examinations during pregnancy, and this is more than medically indicated. At what time in pregnancy is scanning recommended? The clinical value of a routine third trimester scan is not documented.	Thank you for this valuable comment. We have now included in the manuscript, the recommendations issued by the Vietnamese Ministry of Health on gestational weeks when the three ultrasound examinations should be undertaken, please see page 6, lines 134-136 "Additionally, three ultrasound examinations are recommended during pregnancy, in gestational week 11-13, 20-24 and 30-32". We fully agree with your comment that a routine late pregnancy ultrasound examination is not recommended in the third trimester. A Cochrane review from 2015 concludes that routine late pregnancy screening has no benefit for the mother or the baby when used in low risk or unselected pregnancies (Bricker L, Medley N, Pratt JJ. Routine ultrasound in late pregnancy (after 24 weeks' gestation). We have also included this reference in the Discussion, please see page 22, lines 452-455. "The Ministry of Health in Vietnam recommends three ultrasound examinations including one examination also in the third trimester. However, a routine late pregnancy ultrasound in unselected or low-risk populations has been reported to have no benefit for mother or baby".
4	In private sector, up to six examinations are performed in Vietnam. I miss a question about health professionals view on the appropriate number of ultrasound examinations in an uncomplicated pregnancy. I am aware that this question was included in another part of the CROCUS study.	Thank you for the comment. Our original plan was to present this result in another, future paper, but as suggested by the reviewer we have now included this data in this manuscript. Please see the Result section, page 16 lines 345-347. "Participants reported a mean number of 5.9 ultrasound examinations to be medically indicated during an uncomplicated pregnancy (obstetricians/gynaecologists: SD 2.7, range 2-15 and midwives: SD 2.6, range 2-20)." Further, we have also included (in this new version of the manuscript) some new analyses

		with multivariable logistic regression to investigate if the new variable “number of ultrasound examinations indicated in a uncomplicated pregnancy” has some association with the other variables. A number of associations were found and we have included these results in the Result section, page 16, lines 350-355: ”The assessment that four ultrasound examinations or more are medically indicated in an uncomplicated pregnancy was associated with higher agreement with the statement “ultrasound is important for expectant parents to bond with their fetus during pregnancy”, compared to those assessing that three ultrasound examinations or fewer are medically indicated in a normal pregnancy (OR 1.61; 95% CI 1.03-2.50, adjusted for health profession and performing ultrasound or not).”
5	How many questionnaires were distributed and what was the response rate?	All disseminated questionnaires (n=890) were completed by eligible participants. The response rate was thus 100%. As Vietnamese society often functions in a very hierarchical way, the health professionals follow the hospital director’s orders. Although we informed all eligible participants that participation was voluntary, no one declined to answer the questionnaire. We have now included the following paragraph in the Method section, page 10, line 233. “No eligible participant declined participation in the study”.
6	Four important main questions are presented, and the manuscript is well structured in accordance with these questions. “What are health professionals’ views of the role of obstetric ultrasound for clinical management of pregnancy? How do health professionals view access to obstetric ultrasound? How do health professionals assess their skills in performing obstetric ultrasound examinations? What do health professionals believe could improve the utilization of obstetric ultrasound?”	Thank you for these positive comments!
7	The introduction and discussion are well written and highlight the important findings. The figures are illustrative.	Thank you! Although the figures are illustrative, two of them have been removed because of the suggestion in item 1 that there were too many tables and figures included in the manuscript.
8	The power calculation is difficult to understand, because it is based on two different questions. Please explain the power calculation better. In accordance with the power calculation 290 obstetricians and 290 midwives should participate in the study. It is not appropriate to include another group not included in the	Thank you for this very important comment! A challenge for us when designing this study was to adequately determine the size of the study sample. In fact, our study intended to investigate truly novel research questions in the Vietnamese setting, and there were no published results available in this research domain. Therefore, we were forced to make assumptions, and base our power calculation on these assumptions.

	power analyses (58 sonographers) even though sonographers perform ultrasound examinations in Vietnam. The sonographers are only 6.6% of participants, and this group confuses the method section and the result presentation. In some statistical analyses they are a separate group and in other analyses they are combined with obstetricians (Table 3). This leads to multiple analyzing, and high risk of statistical type 1 error. Sonographers are sometimes called physicians (page 11, line 252) and in other sections doctors are called physicians (page 24, line 513). I suggest omitting this group in accordance with the power calculation.	We consulted a statistician and used one outcome question and one background variable and this is described in the section called Sampling, page 9, lines 194-200. For your information, we performed several power calculations with different exposure and outcome variables. Finally, we decided to use the two variables that required the largest sample, i.e. "every woman should undergo ultrasound examination in pregnancy to determine gestational age" in relation to the background variable "work experience over and under 5 years". We agree that this clearly means some uncertainty in relation to the size of the study sample. However, our results indicate that the sample size was probably adequate for investigating our questions of interest. As suggested, we have excluded the sonographers from the study sample and the analyses of this manuscript. We have also included a text about the limitation of the power calculation, please see the section "Strengths and limitations", page 24, lines 532-536. "Since there is a lack of previous studies within this research domain, the power calculation was based on assumptions of proportions for one outcome variable in relation to one background variable. The assumptions in the power calculations therefore means uncertainty of the required size of the study sample. However, the results indicate that the sample size was probably adequate for investigating our questions of interest."
9	Why did you include 535 midwives when the power calculation showed 290 needed? The unequal distribution between doctors and midwives is a limitation which may have influenced results. Please discuss.	Thank you for this comment. As described in item no 8, there were no previous studies available within this research field, and the power calculation was therefore based on assumptions. After deciding how many health facilities we had to include in the data collection to be able to get the calculated sample, we agreed to include all eligible health professionals working the day of the data collection at each health facility. This included a greater number of midwives than needed in the assessment of the power calculation. However, we believe that this was a correct decision since the power calculation was based on assumptions and not on previous results from other studies. As suggested, we have now included a discussion about the sample size and the health professional categories in the section called "Strengths and limitations", please see page 24, line 528-536. "Another limitation might be the unequal distribution between the categories of physicians and midwives in this sample. Although we aimed for equal numbers of each category of health professional, as assessed in the power calculation, all eligible physicians and midwives working the day of data collection were included in the study. This resulted in a higher number of midwives. Since there is a lack of previous studies

		within this research domain, the power calculation was based on assumptions of proportions for one outcome variable together with one background variable. The assumptions in the power calculations therefore mean uncertainty of the required study sample. However, the results indicate that the sample size was probably adequate for investigating our questions of interest.”
10	In Table 3, four degrees of agreement are tested in three groups; leading to a cross table with 12 squares. It is still possible to use chi-square test, but not very interesting. The significance test only shows that differences between the extreme values exist. It is not appropriate to first analyse all three groups separately, and thereafter include sonographers in the group with obstetricians.	Thank you for the comment. As previously suggested we have excluded sonographers from the analyses. All analyses are therefore recalculated and the numbers are updated.
11	The logistic regression analyses are not explained in the statistic section. Which variables were included, and how were they selected?	Thank you for the comment. We have added a description about the variables that were included in the logistic regression analyses, please see the section called Statistical analysis, page 13-14, lines 298-302: ”The independent and dependent variables used for logistic regression are reported in their specific sections above as well as in Table 1. All independent variables were entered into the univariable logistic regression analysis, however only the statistically significant variables were included in the final multivariable logistic regression models.”
12	Limitations related to including the small group of sonographers with all problems related to this are not adequately discussed.	As previously suggested in item 8, the category of sonographers has been excluded from the sample/analyses.
13	The conclusion should only present results from the study and not the authors meanings. The participants are not asked about guidelines, thus, the last sentence in conclusion should be omitted: “In addition, issuing medical guidelines stating clear indications for ultrasound surveillance during pregnancy may contribute to adequate allocation of resources within the health care system in Vietnam.”	Thank you for the comment. As suggested, the last sentence has been removed from the Conclusion section.
14	This study has a very high rate of “self-citing” (11/47 =23%). Please search carefully for relevant references from other groups.	The high number of “self-citing” in the reference list is mainly related to referencing the development of the questionnaire which is the key tool for this study. The items included in the questionnaire were based on the results from the earlier qualitative studies within the CROCUS Study. The CROCUS Study is a two-phase research project, including an initial qualitative phase and a subsequent quantitative phase. The

		questionnaire could not adequately have been developed without the previous qualitative studies. Please see page 9, line 207-208: “The study questionnaire was developed based on the results from the earlier qualitative studies performed in the CROCUS study^{8 20-28}.” We believe that it is important to refer to these publications when explaining the development of the questionnaire, so the readers have the opportunity to access the results in the previous qualitative studies. For other references from the CROCUS Study in the manuscript: There are few studies from other authors within this research field. Therefore, we have used some of the other CROCUS publications as references since there has not been any other option. Taking the above information into account, if the reviewer or the editor still believe that it’s necessary to remove references to the CROCUS Study, we are willing to do so. Please then advise us on which references should be removed.
Reviewer no 2		
Item	Reviewers comments	Authors’ responses
1	The article evaluated the views and experience of health professionals in obstetric ultrasound in Vietnam, the article was good written, long with many tables and figures and discussed briefly the use of obstetric ultrasound as regard the limitations and how to improve the utility of ultrasound but some data needs to be explained. The authors mentioned that some operators may perform 100 exam a day, which mean that the exam time could be <5 min if they work 8 hours continuously, Which is not enough to obtain useful data and the work overload is a factor that limits the proper use of ultrasound.	Thank you for your overall positive comments! Regarding the comment about the number of examinations per day by operators, we agree that the work overload for physicians in Vietnam is substantial. In the literature, other studies have reported that high workload can lead to inadequate information to patients about the ultrasound result and this has been discussed in this paper, please see pages 21, lines 452-456. “It is known that Vietnamese women generally undergo many ultrasound examinations during pregnancy¹⁴, and health professionals performing ultrasound in our study also reported performing up to 100 scans per day per ultrasound operator. High workload can lead to inadequate provision of information to women by the physician, including about indication for ultrasound and the results of the ultrasound examination.”
2	Did the patients receive ultrasound reports?	In the questionnaire there was no item included on whether the patients received ultrasound reports after an examination, therefore it is not possible for us to respond to this question.
3	Did the operators have a certificate of practice ultrasound? Did they receive a qualified ultrasound courses? And what are the requirements that should be fulfilled in the courses? As regards to time, number of exams that they should attend and the ultrasound skills that they acquire?	Thank you for these questions. Unfortunately, we did not ask what kind of ultrasound training the participants had experienced, nor what requirements should be fulfilled in the course or number of scans they should attend. We acknowledge that this is a limitation and we have included this in the section called Strengths and limitations, page 24, lines 536-538.

		“There was no question in the questionnaire about whether the participant had any formal ultrasound training and this may be considered a limitation”.
4	Did the operators learn transvaginal ultrasound to assess early pregnancy and its complications ?.	This question was not included as an aim of our study and can therefore not be answered in our findings.
5	Did the radiologists have a role in obstetric ultrasound?	Yes, the radiologists/sonographers have a role in obstetric ultrasound and perform obstetric ultrasound. However, as suggested by reviewer 1 in item 8, we have now excluded the sonographers from our sample.
6	Did the operators learn when to consult and for whom they refer difficult cases?	Please see item no 4.
7	Why the authors asked the views and experience of many midwives while they lack experience with ultrasound.	Thank you for this comment. In many settings midwives are the main providers of antenatal care, and they often encounter women who have questions about obstetric ultrasound and sometimes they also refer pregnant women for ultrasound examinations. In some countries, for example in Sweden, midwives are the main ultrasound providers of dating ultrasound. The questionnaire used for this study was designed for use in all six countries included in the CROCUS Study, although the healthcare system differs between countries, and health professionals may have different roles and responsibilities in the participating countries. The CROCUS Study data collection has been completed in all participating six countries, and the same questionnaire was used (although in different languages). We also believe that there is a gap in the literature on midwives’ views about obstetric ultrasound, and this study therefore intends to fill that knowledge gap.
Reviewer no 3		
Item	Reviewers comments	Authors’ responses
1	Although the manuscript is well written and complete (as supported by my review checklist), I am of the opinion that the research question and findings add nothing new to the current literature on the specific research topic. The utility of ultrasound in low resource settings have been published umpteenth times including the barriers to the implementation of ultrasound identified in this manuscript.	Thank you for your comment that the manuscript is well written and complete. Unfortunately, we do not agree with the comments by the reviewer on the content/focus of our study. Instead, we believe that our study is presenting novel findings. To the best of our knowledge, there are no previous publications from the Vietnamese setting on the research questions presented in our manuscript. It is correct that there are a number of previous publications on implementation of ultrasound in low resource countries. But as mentioned, this was not the focus of our study. This study is performed in a low to middle resource country where obstetric ultrasound is well-integrated in the maternal health care system.

	Nor was implementation of obstetric ultrasound, or barriers to implementation of ultrasound, the focus of our study. There is a significant lack of epidemiological studies about obstetric ultrasound in Vietnam. When searching on PubMed for “Vietnam and Obstetric Ultrasound”, six paper could be identified and three of them are actually from the CROCUS Study. To be mentioned also is that one of our publications from the qualitative CROCUS Study in Vietnam has been used as a reference in the “Viet Nam Midwifery Report 2016” by UNFPA and Ministry of Health, Vietnam, showing that the results from our project are considered relevant and important.
--	--

VERSION 2 – REVIEW

REVIEWER	Torbjørn Moe Eggebø Trondheim University Hospital, Trondheim Norway
REVIEW RETURNED	21-May-2019

GENERAL COMMENTS	The manuscript has improved, and the authors have responded very well to the criticisms raised. I have no more comments
--

REVIEWER	Birgitte Bruun Nielsen Obstetric Clinic Rigshospitalet Copenhagen University Hospital Copenhagen
REVIEW RETURNED	22-Jul-2019

GENERAL COMMENTS	Thank you for letting me review this paper. This paper addresses a very important issue in a middle income country, here Vietnam. Unfortunately, the authors don't manage to present important news that would be of interest to governments or health planners. In the background the authors mention the importance of the private health sector especially in obstetric ultrasound examinations. Also the important issue of limited salaries that may create incentives for creating additional income through provision of these services. In the last paragraph in the background the authors state that six ultrasound scans are a high number from an international perspective. Yes, but more important it is the double of what is recommended by the Vietnamese government. All this is very important, but unfortunately not addressed in the research. Sampling: Did the authors use purposive sampling or stratified sampling? I don't understand the power calculation. Participants. Please argue why the radiology technicians and sonographers were excluded from the primary sample.
---

	Why were so many midwives included when the power calculation suggested much less? It seems like there was no firm control over the data collection. To answer the research questions I think that the authors would get much more useful information if they had planned their study in a different way. I would suggest in-depth interviews with a series of informants selected based on purposive sampling. The statistical analysis does not fold out and interpret the data so that the reader gets enlightened. Why all the comparison between different groups? It is not the aim to compare health professionals views on this and that. And the results seem obvious and not really interesting. Why did the authors not include the training and supervision of health professionals that perform obstetric ultrasound? The study would be much more interesting if the authors had included the private/public dimension, skills, supervision, prioritization of resources in health, health outcome of ultrasound scans.
--	---

VERSION 2 – AUTHOR RESPONSE

Reviewer no 1		
Item	Reviewers comments	Authors' responses
1	The manuscript has improved, and the authors have responded very well to the criticisms raised. I have no more comments.	Thank you very much for your positive evaluation, and that you are content with the current version of the manuscript.
Reviewer no 2		
1	Thank you for letting me review this paper. This paper addresses a very important issue in a middle income country, here Vietnam.	Thank you, we also believe that this is a very important topic.
2	Unfortunately, the authors don't manage to present important news that would be of interest to governments or health planners.	Unfortunately, we disagree with the reviewer's comment that the novelties in this study would not be of interest to governments and health planners. We believe that our results are interesting and valuable for health professionals as well as for policy makers and health planners. The need for additional training for ultrasound operators, as this study reports, may be valuable for health planners especially since there is a lack of previous studies examining this topic in the Vietnamese context. To the best of our knowledge, the Vietnamese Ministry of Health struggles to manage overcrowded hospitals. We believe that the overprovision of ultrasound examinations in normal pregnancy needs to be further discussed, and that this study also indicates that ultrasound operators need to be part of this discussion and subsequent adequate measures.
3	In the background the authors mention the importance of the private health sector especially in obstetric ultrasound examinations.	For clarification concerning the private health sector, our assumption was that participants working at public health facilities more often perform ultrasound examinations that are based on medical indications

		than those working at private health facilities because of other priorities and incentives. We therefore focused on public health facilities, as we were interested of for example access to obstetric ultrasound for all pregnant women in the Hanoi area, not only those who can afford to pay for their examinations.
4	Also the important issue of limited salaries that may create incentives for creating additional income through provision of these services.	We agree that it would be interesting to also investigate Vietnamese obstetricians/gynaecologists work environment including incentives for additional income related to private ultrasound examinations. However, this was not the aim of this study, and it was therefore not investigated.
5	In the last paragraph in the background the authors state that six ultrasound scans are a high number from an international perspective. Yes, but more important it is the double of what is recommended by the Vietnamese government. All this is very important, but unfortunately not addressed in the research.	Regarding the number of ultrasound examinations that pregnant women in Vietnam undergo, we agree that it is interesting as a few other studies indicate that it strongly deviates from the national guidelines of the recommended number of ultrasound examinations. We therefore asked the participants, through the questionnaire, about the number of ultrasound examinations that they considered medically indicated in an uncomplicated pregnancy. These results are presented in the “Results” section called “The role of obstetric ultrasound”, page 16, lines 349-351. “Participants reported an average number of 5.9 ultrasound examinations as medically indicated during an uncomplicated pregnancy (obstetricians/gynaecologists: SD 2.7, range 2-15, and midwives: SD 2.6, range 2-20).” We have followed your suggestions, and we have performed some new calculations of the number of ultrasound operators that align with the national guidelines, i.e. the recommendations of three ultrasound examinations in a normal pregnancy. Thereafter, we have added the following text in the “Results” section called “The role of obstetric ultrasound, please see page 16, lines 351-353” “A quarter of the ultrasound operators (25.1%) agreed with the national guidelines that three ultrasound examinations are medically indicated during an uncomplicated pregnancy.” We have also inserted new text in the “Discussion” section, please see page 22, lines 437-439. “Participants reported an average of almost six ultrasound examinations as medically indicated in an uncomplicated pregnancy in contrast to the three ultrasound examinations that are recommended in the Vietnamese national guidelines.” Please see also the following lines in the “Discussion” section, page 23, lines 474-481. “Although it is well known that ultrasound examinations hold a strong appeal for pregnant women, the high number of examinations that pregnant women receive seems also to be supported by the ultrasound operators, as the participants in this study considered that twice as many ultrasound examinations were medically indicated compared to

		the recommendations by the Ministry of Health, Vietnam. Further, participants' report of a mean number of almost six ultrasound examinations to be medically indicated is also in great contrast to the Cochrane review supporting the 2016 WHO ANC guideline recommending a single ultrasound examination before 24 weeks of gestation" In addition to the number of examinations reported by the reference in the background section (Gammeltoft T. Nguyen HT. The commodification of obstetric ultrasound scanning in Hanoi, Viet Nam), we have not been able to find any official data/statistics on the actual number of ultrasound examinations that pregnant women receive in Vietnam. However, we have added another reference (Tran et al. Urban-rural disparities in antenatal care utilization: a study of two cohorts of pregnant women in Vietnam) that also report a mean value of six ultrasound examination for urban pregnant women. Please see the "Background" section, page 7, line 146 and the "References", page 30, lines 666-668, Reference no 14. In conclusion, the number of ultrasound examinations that pregnant women receive need to be further investigated in some other study.
6	Sampling: Did the authors use purposive sampling or stratified sampling?	Thank you for this question. Regarding the sampling, we have described that purposive sampling was used in the section called "Sampling", page 9, lines 202-205. "Purposive sampling was used to obtain a representative sample of health professionals caring for pregnant women at different levels of health facilities in urban, semi-urban and rural areas in the region of Hanoi. One national hospital, one provincial hospital, 24 district hospitals and three maternity homes were included in the study, in total 29 health facilities."
7	I don't understand the power calculation.	Thank you for your comment on the need of clarification of the power calculation. To perform an adequate power calculation for this study was a substantial challenge, since there were no previous published results from this research domain and/or from this setting. It was therefore difficult to determine the adequate size of the study sample. Consequently, we were obliged to make tentative assumptions and base our power calculation on these assumptions. We consulted a statistician to perform the power calculation. We performed several power calculations with different exposure and outcome variables. Finally, we used one background variable (exposure) and one outcome variable that would require the largest sample in order to reach a sufficient sample size. As the outcome variable we chose the statement "every woman should undergo ultrasound examination in pregnancy to determine gestational

		age". As exposure we used the variable categorizing obstetricians/gynaecologists that had either worked less than five years or had worked five years or more. To detect a difference in opinion with a proportion of 0.10 i.e. 10% difference in opinion between the two groups of obstetricians/gynaecologists, and a power of 80% and a significance level of 5%, we needed 290 participants. The same procedure was done for the group of midwives. This is described in the section called "Sampling", page 9, lines 195-201. "Owing to the lack of findings from similar studies, a sample size of 290 obstetricians/gynaecologists and a corresponding number of midwives (n=290) was calculated based on plausible estimations of prevalence of background characteristics and outcome variables. The calculation was based on the outcome requiring the largest sample size "every woman should undergo ultrasound examination in pregnancy to determine gestational age" and the background variable "work experience over and under 5 years", to detect a difference in proportion of 0.10 with a power of 80% and a significance level of 5%." A new text in the "Discussion" section about the limitation of the power calculation was added in the previous version of the manuscript (i.e. previous revision). Please see the section "Strengths and limitations", page 27, lines 560-564. "Since there is a lack of previous studies within this research domain, the power calculation was based on assumptions of proportions for one outcome variable in relation to one background variable. The assumptions in the power calculations therefore means uncertainty of the required size of the study sample. However, the results indicate that the sample size was probably adequate for investigating our questions of interest."
8	Participants: Please argue why the radiology technicians and sonographers were excluded from the primary sample.	Thank you for your comment. The plain answer to your question is that the exclusion was a request by reviewer no 1 in his previous comments on this manuscript. Thus, we complied to his suggestion. Accordingly, we excluded the sonographers from the study sample in our revised version of the manuscript. The radiology technicians did not fulfil the inclusion criteria for participation in the study, and were excluded accordingly. To be mentioned is that the sonographers constituted only 6.6% of the participants. Reviewer no 1 argued that the category of sonographers confused the manuscript, implying difficulties in the presentation of the study/data in the Methods section and the Results section. For clarification purposes, we have now added one additional sentence in the manuscript why the group

		of sonographers and radiology technicians were excluded from the final sample. Please see the section called “Data collection procedures including recruitment of study participants”, page 10, lines 235-237. “ Six individuals working as radiology technicians were excluded from the primary sample as the they did not fulfil the inclusion criteria, and 60 sonographers were also excluded from the primary sample as suggested by the journal.”
9	Why were so many midwives included when the power calculation suggested much less? It seems like there was no firm control over the data collection.	Thank you for this question and comment. Regarding the question related to the final (greater) number of midwives included in the study, compared to the final number of obstetricians /gynaecologists included in this study; it is related to the uncertainty of the sufficient size of the study sample. The uncertainty of the basis for the power calculation has been described in item no 7 above, as well as in the previous revised manuscript and the response to reviewer no 1 (see attached document). As earlier described there were no previous studies available within this research field, and the power calculation was therefore based on assumptions (see response to item 7 above). After deciding how many health facilities we needed to include in the data collection, to be able to reach the calculated sample size, we agreed to include all eligible health professionals working the day of the data collection at each health facility. It was in fact impossible for us to get information in advance of the data collection, how many midwives that would be on duty and available/eligible to participate in the study during their work shift. In order to avoid introducing a selection bias, we decided to invite all available midwives to participate in the study. In fact, this resulted in a greater number of midwives’ participants than needed in relation to the estimation in the power calculation. However, we believe that this was a correct decision since the power calculation was based on tentative assumptions and not on published results, and it was therefore uncertain. To be mentioned is that performing data collection in low-to-middle income countries includes several challenges, as for example the uncertainty of available number of participants at the data collection in the field. In the last, revised version of this manuscript, a reasoning section was added about the sample size and the health professional categories in the section called “Strengths and limitations”, please see pages 26-27, line 556-560. “Another limitation might be the unequal distribution between the categories of physicians and midwives in this sample. Although we aimed for equal numbers of

		each category of health professional, as assessed in the power calculation, all eligible physicians and midwives working the day of data collection were included in the study. This resulted in a higher number of midwives.” Regarding your comment that it seemed to be no firm control over the data collection, we would like to inform you that there was a DAILY contact between the first author and the data collectors on the number of participants included in the study. We therefore consider that the data collection procedures were adequately surveyed.
10	To answer the research questions I think that the authors would get much more useful information if they had planned their study in a different way. I would suggest in-depth interviews with a series of informants selected based on purposive sampling.	Thank you for this comment and we agree on the importance of qualitative research in this scientific area. Your suggestion of a procedure is actually what we undertook as the first phase of the international Cross Country UltraSound study (CROCUS study), see the 11 publications listed below this table. The results of the in-depth interviews with obstetricians /physicians and the focus group with midwives were the basis of the composition of the questionnaire used in this quantitative study.
11	The statistical analysis does not fold out and interpret the data so that the reader gets enlightened. Why all the comparison between different groups? It is not the aim to compare health professionals views on this and that. And the results seem obvious and not really interesting	We would like to clarify that the whole sample was used at the initial analyses of the selected variables. Thereafter we performed analyses related to sub-categories/health professionals. A number of significant differences were demonstrated which we assess are of interest scientifically. For example, to improve the utilisation of ultrasound during pregnancy and regulations regarding it’s use in the Vietnamese context, we believe it is important to investigate the different opinions among the categories of health professionals working within maternity care in Vietnam. Although we have already reported obstetricians/ gynaecologists’ and midwives’ views of ultrasound in the qualitative studies (see the references below), the findings in this study provide quantification of different views/experiences by health professionals which we consider are of value. In summary, we disagree with the reviewer’s comment that the results seem obvious and not interesting. We believe that our results are interesting and valuable for health professionals as well as for policy makers. Further, our survey in Vietnam includes data on a number of (unpublished) interesting items in areas where there are no previous scientific publications.
12	Why did the authors not include the training and supervision of health professionals that perform obstetric ultrasound?	Thank you for this question. This is unfortunately an omission from our side when we composed the questionnaire.

		However, we would have encountered some difficulties in formulating these items in order to be able to make comparisons between countries. Comparisons between the different countries have always been an underlying objective of the international CROCUS Study. We acknowledge that this is a limitation. As this question was also raised by reviewer no 2 in the first revision round, we included some paragraphs in that revised version of the manuscript in the section called "Strengths and limitations", page 27, lines 564-566. "There was no question in the questionnaire about whether the participant had received any formal ultrasound training and this may be considered a limitation."
13	The study would be much more interesting if the authors had included the private/public dimension, skills, supervision, prioritization of resources in health, health outcome of ultrasound scans.	Thank you for your comments. We agree that that your suggestions on additional research topics are interesting and can constitute future research projects.  • As discussed in our response to item no 3, we did not aim to include private health facilities in this study. • Regarding the skills of the ultrasound performers, one of our research questions were "How do health professionals assess their skills in performing obstetric ultrasound examinations?". We believe that this study answers this question. However, ultrasound skills can also be assessed by an external part but this was not the aim of this study, and would have required a completely different research approach. • Supervision of ultrasound operators is an important aspect, but was not a part of this study. • Prioritisation of resources of health is an interesting topic but for this study we focused on how to improve utilisation of ultrasound. For your information, prioritisation of resources of health within maternity care was discussed in another of our manuscripts, i.e. the qualitative CROCUS study with midwives in Vietnam (a submitted manuscript that is under current revision). • Regarding the health outcome of ultrasound scans, several studies have already investigated this topic, please see for example Whitworth M, Bricker L, Mullan C. Ultrasound for fetal assessment in early pregnancy. Cochrane Database Syst Rev 2015(7):CD007058 and Bricker L, Medley N, Pratt JJ. Routine ultrasound in late pregnancy (after 24 weeks' gestation). Cochrane Database of Systematic Reviews 2015(6). This was therefore not the aim of this study although a very important topic within maternity care.

Qualitative publications in the Cross Country UltraSound study (CROCUS):

- Edvardsson K, Small R, Persson M, Lalos A, Mogren I. "Ultrasound is an invaluable third eye, bit it can't see everything": a qualitative study with obstetricians in Australia. *BMC Pregnancy Childbirth*. 2014 Oct 22;14:363. doi: 10.1186/1471-2393-14-363.
- Edvardsson K, Small S, Lalos A, Persson M, Mogren I. Ultrasound's window on the womb brings ethical challenges for balancing maternal and fetal health interests: obstetricians experiences in Australia. *BMC Pregnancy and Childbirth*. *BMC Medical Ethics*.2015, 16:31DOI: 10.1186/s12910-015-0023-y.
- Edvardsson K, Mogren I, Lalos A, Persson M, Small R. A routine tool with far-reaching influence: Australian midwives' views on the use of ultrasound during pregnancy. *BMC Pregnancy and Childbirth*. *BMC Pregnancy Childbirth*. 2015 Aug 27;15:195. doi: 10.1186/s12884-015-0632-y.
- Edvardsson K, Graner S, Pham Thi Lan, Åhman A, Small S, Lalos A, Mogren I. "Women think pregnancy management means obstetric ultrasound": Vietnamese obstetricians' views on the use of ultrasound during pregnancy. *Glob Health Action*. 2015 Oct 29;8:28405. doi: 10.3402/gha.v8.28405. eCollection 2015.
- Åhman A, Persson M, Edvardsson K, Lalos A, Graner S, Small R, Mogren I. Two sides of the same coin – an interview study of Swedish obstetricians' experiences with ultrasound during pregnancy. *BMC Pregnancy Childbirth*. 2015 Nov 20;15(1):304. doi: 10.1186/s12884-015-0743-5.
- Edvardsson K, Ntaganira J, Åhman A, Semasaka Sengoma JP, Small R, Mogren I. Physicians' experiences and views on the role of obstetric ultrasound in rural and urban Rwanda: a qualitative study. *Trop Med Int Health*. 2016 Jul;21(7):895-906.
- Åhman A, Kidanto HL, Nagarina M, Edvardsson K, Small R, Mogren I. "Essential but not always available when needed" –an interview study of physicians' experiences and views regarding use of obstetric ultrasound in Tanzania. *Global Health Action*. 2016 Jan;9(1):31062.
- Edvardsson K, Lalos A, Annika Åhman, Small R, Graner S, Mogren I. Increasing possibilities – increasing dilemmas: A qualitative study of Swedish midwives'. *Midwifery*. 2016 Nov;42:46-53.
- Holmlund S, Ntaganira J, Edvardsson E, Pham Thi Lan, Semasaka S JP, Åhman A, Small R, Mogren I. "Improved maternity care if midwives learn to perform ultrasound": a qualitative study of Rwandan midwives' views and experiences of obstetric ultrasound. and views of obstetric ultrasound. *Global Health Action*, 10:1, 1350451, DOI: 10.1080/16549716.2017.1350451
- Åhman A, Edvardsson K, Kidanto HL, Nagarina M, Small R, Mogren I. Without ultrasound you can't reach the best decision – Midwives' views and experiences of the role of ultrasound in hospital care in Dar Es Salaam, Tanzania. *Sex Reprod Healthc*. 2018 Mar;15:28-34. doi: 10.1016/j.srhc.2017.11.007. Epub 2017 Nov 22.
- Edvardsson K, Åhman A, Fagerli TA, Darj E, Holmlund S, Small R, Mogren I. Norwegian obstetricians' experiences of the use of ultrasound in pregnancy management. A qualitative study. *Sexual & Reproductive Healthcare*. *Sex Reprod Healthc*. 2018 Mar;15:69-76. doi: 1